# Decoding Pediatric Laryngopharyngeal Reflux: Unraveling Symptoms Through Multichannel Intraluminal Impedance and pH Insights

**DOI:** 10.3390/diagnostics15010034

**Published:** 2024-12-26

**Authors:** Ivan Pavić, Petar Prcela, Josip Pejić, Irena Babić, Ana Močić Pavić, Iva Hojsak

**Affiliations:** 1Department of Pulmonology, Allergology and Immunology, Children’s Hospital Zagreb, 10000 Zagreb, Croatia; 2School of Medicine, University of Split, 21000 Split, Croatia; prcelapetar@gmail.com; 3Department of Thoracic Surgery, Clinical Hospital Dubrava, 10000 Zagreb, Croatia; jpejic30@gmail.com; 4Otorhinolaryngology Department, Children’s Hospital Zagreb, 10000 Zagreb, Croatia; irena.babic4@gmail.com; 5Referral Center for Pediatric Gastroenterology and Nutrition, Children’s Hospital Zagreb, 10000 Zagreb, Croatia; amocicpavic@gmail.com (A.M.P.); ivahojsak@gmail.com (I.H.); 6School of Medicine, University of Zagreb, 10000 Zagreb, Croatia; 7School of Medicine Osijek, University Josip Juraj Strossmayer, 31000 Osijek, Croatia

**Keywords:** laryngopharyngeal reflux, multichannel intraluminal impedance-pH monitoring, children, symptoms, fiberoptic ENT findings

## Abstract

**Background:** The relationship between symptoms of laryngopharyngeal reflux (LPR) and objective reflux measurements obtained through multichannel intraluminal impedance-pH (MII-pH) monitoring remains unclear. **Objectives:** The aim of this study was to investigate the relationship between LPR symptoms and objective reflux episodes and possible associations between fibreoptic ENT findings, eosinophil counts, and serum IgE levels with reflux episodes detected by MII-pH. **Methods:** In this prospective study, MII-pH monitoring, fiberoptic laryngoscopy, nasal swabs for eosinophils, total serum IgE levels, and symptom assessment (Reflux Symptom Index, RSI) were performed in all children with suspected LPR. The Reflux Findings Score (RFS) was determined based on the laryngoscopy findings. **Results:** A total of 113 patients (mean age, 8 years) with LPR symptoms were included in the study. The number of reflux episodes was highest in children with chronic cough and recurrent broncho-obstruction. Secondary outcomes showed positive correlations between reflux episodes and ENT findings, particularly hypopharyngeal hyperemia, arytenoid hyperemia, and arytenoid erythema (*p* < 0.01, *p* < 0.001, and *p* < 0.001, respectively). The number of total, acidic, and weakly acidic reflux episodes was significantly positively correlated with RSI and RFS. Proximal total, acidic, and weakly acidic reflux episodes showed significant correlations with eosinophil counts in nasal swabs but negative correlations with serum IgE levels. **Conclusions:** This study highlights the significant role of weakly acidic reflux in pediatric LPR and its association with respiratory symptoms. Our findings emphasize the importance of objective monitoring techniques in the assessment of LPR and provide insights for refining diagnostic and management strategies.

## 1. Introduction

Laryngopharyngeal reflux (LPR), characterized by the retrograde flow of gastric contents into the upper aerodigestive tract, presents a diagnostic challenge in the pediatric population due to its diverse clinical manifestations and the lack of a definitive gold standard for diagnosis [1]. While the prevalence of gastroesophageal reflux (GER) in children is well established, understanding the nuances of LPR, particularly its impact on the airway, remains an evolving area of research.

The spectrum of LPR-related symptoms in children ranges from respiratory symptoms, such as chronic cough and bronchial obstruction, to otolaryngological manifestations, such as throat clearing, otitis media with effusion (OME), and globus sensation [2]. The identification and characterization of LPR as a potential etiological factor for these symptoms have led to further investigation of their pathophysiology and clinical implications. The complex interplay between these symptoms and objective measures of GER requires a comprehensive approach for accurate diagnosis and tailored management.

Studies have shown that the diagnosis of LPR remains difficult, mainly due to the lack of specific symptoms and reliable diagnostic methods [3]. Symptom-based assessments, such as the Reflux Symptom Index (RSI), provide valuable insights but are limited by subjectivity and the overlap of symptoms with other conditions.

Recent advances in diagnostics, such as multichannel intraluminal impedance and pH (MII-pH), offer the possibility of investigating the features of GER in children with suspected LPR [4]. This technique allows differentiation between acidic, weakly acidic, and non-acid reflux episodes, contributing to a better understanding of reflux patterns and their potential impact on the laryngopharyngeal mucosa [1].

To date, numerous studies have investigated the relationship between LPR symptoms and GER using MII-pH monitoring [5,6,7,8,9,10,11,12]. For example, Thilmany and colleagues reported the prevalence of proximal acid reflux in children with chronic pulmonary symptoms [13]. In addition, Ghezzi and colleagues emphasized the importance of weak acid reflux in children with difficult-to-treat respiratory symptoms [14].

However, the correlation between GER episodes characterized by MII-pH monitoring and clinical findings such as fiberoptic laryngoscopy and inflammatory markers remains a subject of ongoing investigation. Despite the existing literature, there are still gaps in our understanding of the interplay between MII-pH findings, clinical symptoms, and other diagnostic parameters associated with LPR.

This study aims to contribute to existing knowledge by characterizing the MII-pH profiles of children with LPR symptoms, correlating these findings with the results of fiberoptic ENT examination, eosinophil counts in nasal swabs, and serum IgE levels, and assessing their relationship with reflux-associated scores. By investigating these relationships, this study aims to provide valuable insights into the diagnostic landscape of pediatric LPR and pave the way for more sophisticated and targeted approaches to patient care. Given the gaps in current knowledge and the need for a holistic understanding of the disease, our research endeavors to advance the field and improve the accuracy of diagnosis and management strategies for children with suspected LPR.

## 2. Materials and Methods

### 2.1. Study Design

This prospective observational study was conducted at the Department of Pediatric Pulmonology, Zagreb Children’s Hospital, between November 2017 and December 2023. This study enrolled pediatric patients with symptoms suggestive of laryngopharyngeal reflux (LPR), such as chronic cough, recurrent bronchial obstruction, throat clearing, otitis media with effusion (OME), and globus sensation. Ethical approval was granted by the Ethics Committee of Zagreb Children’s Hospital (protocol number: 02-26/10-5-17), and written informed consent was obtained from all participants’ parents or legal guardians prior to study enrolment.

### 2.2. Participants

Children aged up to 18 years with persistent LPR-related symptoms lasting at least three months were included. The primary inclusion criteria were respiratory and otolaryngological symptoms with no proven etiology and suspected to be related to LPR as one of the possible causes of unexplained symptoms. Each child underwent a thorough evaluation that included a detailed medical history and comprehensive physical examination. Pulmonary function tests were conducted in children who were able to cooperate during testing to assess respiratory capacity and functionality. In addition to these tests, a series of hematologic, radiologic, immunologic, and allergy assessments were performed to systematically rule out other potential underlying diseases or conditions contributing to their symptoms. Special attention was given to the role of environmental factors as potential contributors to children’s symptoms. A detailed environmental exposure history was obtained, focusing on identifying factors, such as exposure to cigarette smoke, indoor air pollution, mold, and other airborne irritants. Families were specifically questioned about smoking habits within the household and frequent exposure to secondhand smoke from other environments. Children with confirmed exposure to these environmental factors were excluded from the final analysis to minimize confounding variables. This exclusion criterion ensured that the study results reflected cases in which symptoms were less likely to have been influenced by modifiable external factors.

The exclusion criteria were as follows:Severe neurological impairmentStructural abnormalities of the airways or upper gastrointestinal tractCongenital craniofacial anomaliesImmunodeficiency disordersSuspected or confirmed asthma or allergic rhinitisRecent use of acid-suppressive medications, prokinetics, or intranasal corticosteroids

Patients underwent a standardized diagnostic protocol that included assessment by a pediatric pulmonologist and an otolaryngologist (ENT specialist), 24-h MII-pH monitoring, nasal swabs to assess eosinophilia, and measurement of total serum IgE levels. The Reflux Symptom Index (RSI) was calculated based on the symptoms reported by the children and/or their parents/caregivers [15].

### 2.3. ENT Examination

Prior to 24 h MII-pH monitoring, each child was thoroughly examined by a single otolaryngologist. In addition to a complete history and physical examination, each child underwent transnasal fiberoptic laryngoscopy using a Karl Storz 11101SK2 Rhino-Laryngoscope Fiberscope (Karl Storz SE & Co. KG, Tuttlingen, Germany) immediately prior to 24 h MII-pH monitoring. The Reflux Finding Score (RFS) was calculated based on fiberoptic laryngoscopy findings, scoring eight items, including edema, erythema, and hyperemia of the laryngeal structures [16]. Mucosal hyperemia, arytenoid erythema, and edema of the cricoid area were noted, with a higher score indicating more severe findings. ENT findings were documented and analyzed in relation to reflux characteristics and other diagnostic markers.

### 2.4. Esophageal MII-pH Monitoring

All patients underwent 24-h ambulatory MII-pH monitoring using an Ohmega MII-pH system (MMS, Enschede, The Netherlands). The catheter used for MII-pH monitoring was selected based on the child’s height and featured an antimony pH sensor with six pairs of impedance electrodes. The catheter was inserted nasally, and the pH sensor was positioned on the third vertebral body above the diaphragmatic angle. The catheter’s position was confirmed by X-ray. Parents were instructed to maintain a detailed diary of the child’s activities, including mealtimes, posture, and the timing of the occurrence of laryngopharyngeal symptoms. The MII-pH recordings were uploaded to a personal computer at the end of the recording time and manually analyzed using the MMS software package (Version 9.6a) according to the criteria of the consensus statement on combined MII-pH monitoring in children [17]. Reflux episodes were classified as acidic (pH < 4), weakly acidic (pH 4–7), or non-acidic (pH ≥ 7). Proximal and distal reflux events were also noted based on their occurrence in the esophagus. As noted in previous studies, a 2-min window following the onset of a reflux episode was arbitrarily selected to determine the association between laryngopharyngeal symptoms and GER [18]. A symptom was classified as associated with GER if it occurred within this 2-min window. Any laryngopharyngeal symptoms occurring outside this timeframe were considered unrelated to the GER. Meal periods were excluded from the analysis to ensure that only postprandial and fasting data were considered in the analysis of GER events. Exclusion was performed based on the timing of food intake, as recorded in the patient’s diary, and verified by the attending clinician.

### 2.5. Nasal Swabs and IgE Levels

Nasal swabs were obtained from each participant to determine the presence of eosinophils, which are considered an indicator of local inflammation. The nasal smear slide for eosinophil granulocytes was stained with Giemsa and examined microscopically by the same cytologist.

Total serum IgE levels were measured using blood samples. Blood samples were collected between 8 am and 12 pm in serum gel tubes. Blood was left to coagulate spontaneously at room temperature (20–24 °C) for 60 to 120 min. The serum was separated by centrifugation at 1000–1300× *g* for 10 min at room temperature. The total IgE was measured in fresh samples. The concentration of total IgE was determined by the standardized fluoroimmunoassay ImmunoCAP-method with monoclonal and polyclonal antibodies (Phadia ImmunoCAP, Thermo Fisher Scientific, Uppsala, Sweden) on a selective UniCAP 100 auto-analyzer (Thermo Fischer, Uppsala, Sweden), using reagents from the same manufacturer. Elevated levels indicate atopic disease, although children diagnosed with allergic rhinitis and asthma were excluded from the study.

### 2.6. Symptom Assessment

The Reflux Symptom Index (RSI) was used to assess the severity and frequency of symptoms reported by the children and/or their parents/caregivers. The RSI is a validated questionnaire that rates nine symptoms on a scale of 0 to 5, with higher scores indicating greater severity of symptoms [15]. The Reflux Finding Score (RFS) was also calculated based on ENT findings, which include visible signs of inflammation and tissue changes attributable to reflux [16].

### 2.7. Outcome Measures

The primary outcome was to determine the MII-pH characteristics (number and type of reflux episodes) in children with suspected LPR based on clinical symptoms and ENT findings.

The secondary outcomes included the following:Correlation of fiberoptic ENT findings with the number of proximal, acidic, and weakly acidic GER episodes.Assessment of the relationship between the number of proximal and distal reflux episodes (acidic, weakly acidic, and non-acidic) and RSI/RFS scores.Correlation between eosinophils in nasal swabs, serum IgE levels, and number of reflux episodes.

By integrating MII-pH profiles, clinical findings, and biochemical markers, this study aimed to provide a comprehensive understanding of the diagnostic landscape of pediatric LPR and to investigate potential biomarkers that could improve diagnostic accuracy and patient care.

### 2.8. Statistical Analysis

All statistical analyses were performed using SPSS version 26.0 (Chicago, IL, USA). Descriptive statistics were calculated for demographic data and clinical variables, with continuous data presented as means and standard deviations or medians and interquartile ranges, depending on the data distribution. Pearson or Spearman correlation tests were used to assess the relationships between reflux characteristics, symptom scores (RSI), clinical findings (RFS), and eosinophil/IgE levels. Differences between symptom groups were analyzed using ANOVA. Statistical significance was set at *p* < 0.05 for all tests.

## 3. Results

### 3.1. Patients

A total of 113 children were included during one study period. The demographic data are shown in Table 1.

Most patients had a chronic cough (*n* = 53, 47%), followed by recurrent bronchial obstruction (*n* = 23, 20%), throat clearing (*n* = 17, 15%), otitis media with effusion (*n* = 10, 9%), globus sensation (*n* = 9, 8%), and recurrent pneumonia (*n* = 1, 1%). Gastrointestinal symptoms occurred in 21 (19%) children. Eosinophils were present in the nasal swabs of 72 (64%) children.

#### 3.1.1. ENT Findings

Figure 1 presents the ENT findings.

The most common condition is arytenoid hyperemia, which is reported in 86% of cases, followed by hyperemia of the hypopharynx in 40%, and adenoid hypertrophy in 31%. Mucosal hyperemia is seen in 32% of cases, while other conditions, such as arytenoid erythema and supraglottic hyperemia, are seen in 28% and 23% of cases, respectively. Less common findings include cricoid edema and Eustachian tube dysfunction (both 7%), while hypertrophy of the lingual tonsils is rare at 2%. The median number of ENT findings per symptom group was 3 (range, 0–8), and there was no difference in the number of ENT findings between symptom groups (*p* = 0.388).

#### 3.1.2. Primary Outcome Measures

The primary outcome measures are presented in Table 2.

The median number of reflux episodes determined by the MII-pH was 63 (range, 57), and the highest number were weakly acid episodes (median 31, range, 36). The differences in the number of reflux episodes between the groups of patients stratified by symptoms are shown in Table 3.

The analysis showed that children with chronic cough and recurrent bronchial obstruction had a significantly higher number of all reflux episodes than children with other symptoms.

#### 3.1.3. Secondary Outcome Measures

The only fibroendoscopic findings that correlated positively with proximal total, acidic, and weakly acidic GER were hypopharyngeal hyperemia (Spearman coefficient 0.388, 0.364, 0.243; *p* < 0.01; respectively), arytenoid hyperemia (Spearman coefficient 0.477, 0.387, 0.429; *p* < 0.001; respectively), and arytenoid erythema (Spearman coefficient 0.461, 0.365, 0.381; *p* < 0.001; respectively).

The correlations between RSI and RFS and the number of reflux episodes are shown in Table 4.

The number of total, acidic, and weakly acidic reflux episodes was significantly positively correlated with RSI and RFS. The number of alkaline, total proximal episodes, proximal acidic, and weakly acidic episodes correlated positively with RFS only.

The correlation between eosinophils in nasal swabs, serum IgE levels, and the number of reflux episodes is shown in Table 5.

The number of reflux episodes, especially acidic and proximal episodes, showed significant positive correlations with eosinophils in the nasal swab (*p* < 0.001 and *p* < 0.001, respectively), indicating a relationship between reflux and nasal inflammation. In contrast, reflux episodes generally showed a negative correlation with IgE levels, with significant correlations found specifically for proximal and proximal acidic episodes (*p* = 0.014 and *p* = 0.038, respectively). This indicates that although reflux correlates with eosinophil activity in the nose, it has a weaker and predominantly negative association with IgE levels.

## 4. Discussion

The primary outcome of our study, which focused on the features of MII-pH in children with suspected LPR, showed a diverse spectrum of symptoms, with chronic cough being the predominant complaint in almost half of the cases. This high prevalence highlights the need for a thorough investigation of respiratory symptoms in pediatric patients, as these could potentially indicate an underlying LPR. The primary outcome analysis revealed that patients with chronic cough and recurrent bronchial obstruction had a higher number of all reflux episodes compared to other symptom groups.

Compared to previous studies, our investigation represents several notable advances in the understanding of LPR in the pediatric population. A study by Greifer and colleagues, which included 63 pediatric participants aged 0 to 21 years, found no association between extraesophageal signs and symptoms and pathological GER based on the DeMeester score or impedance testing [19]. The most common indication in this study was cough/asthma. Only 10 of the 63 patients had an abnormal score based on impedance criteria and 46 patients had a symptom index, with only 15.2% showing a positive correlation between symptoms and reflux events [19]. In a recently published study by Mantegazza and colleagues, which included 197 pediatric participants, infants and children with a mean age of 8 years were included [10]. The study found no significant correlation between the RSI or RFS and the MII-pH parameters. In addition, the sensitivity and specificity of RSI and RFS for identifying abnormal MII-pH results were low, suggesting that the use of these scores alone for the diagnosis of LPR in pediatric patients is limited. The studies differed in their findings. While our study found positive associations between RSI and RFS scores and the frequency of episodes of total and proximal acid and weak acid reflux, a second study found no significant differences in MII-pH parameters based on RSI or RFS scores [10]. The study by Eiamkulbutr and colleagues investigated the prevalence of GERD in children with extraesophageal symptoms using a combined video and MII-pH approach. In 51 children with symptoms such as cough and recurrent pneumonia, GERD was detected in 35.3% of the cases in the study, with video surveillance increasing the diagnostic yield [20]. The study highlights the longest reflux time and mean nocturnal baseline impedance as valuable parameters and suggests that these metrics could improve the diagnostic criteria for pediatric GERD. The discrepancies in the detection of positive GER between our study and that of Eiamkulbutr and colleagues are likely due to differences in patient symptoms, diagnostic parameters, and monitoring methods. Our study focused on children with LPR symptoms and specific ENT findings, emphasizing weakly acidic reflux, whereas Eiamkulbutr and colleagues studied children with extraesophageal GERD symptoms and used novel parameters, such as longest reflux time and combined video and MII-pH monitoring. These differences in study focus and diagnostic tools contribute to the differences in GER positivity between the two studies.

The results of our study are consistent with some previously published data [7,13,21,22,23]. In a retrospective review, Sacco and colleagues investigated airway inflammation in children with predominantly weak acid reflux or acid reflux causing respiratory symptoms [22]. They found a similar number of cells harvested by bronchoalveolar lavage (BAL) in both groups, but observed a higher proportion of BAL epithelial cells in children with weakly acidic reflux, indicating greater airway damage [22]. A significant correlation was found between the number of weakly acidic reflux events and the proportion of BAL epithelial cells. These results highlight the significant impact of weak acid reflux on airway inflammation and damage in children and suggest that conventional anti-acid treatments may not fully address the associated pathophysiological processes. Another study included 104 children with suggestive LPR symptoms, with a mean age of 8.9 years, and showed a significant association between these symptoms and both acid and weakly acid reflux episodes detected by MII-pH monitoring [23]. Our findings are consistent with those of a larger study involving 426 children with chronic respiratory symptoms, which also showed a significant association between GER and respiratory symptoms, particularly chronic cough [24]. In both studies, weakly acidic events were found to be common and associated with certain symptom profiles, such as chronic cough. The agreement between our findings and those observed in the larger cohort strengthens the evidence base for the use of MII-pH monitoring to assess GER in children with chronic respiratory and ENT symptoms.

In a prospective pilot study, Mahoney LB and colleagues performed non-targeted global metabolomic profiling of BAL fluid from 43 children undergoing evaluation for chronic respiratory symptoms with bronchoscopy, upper endoscopy, and pH-MII monitoring [25]. The study found significantly lower concentrations of histamine, malate, adenosine 5’-monophosphate, and ascorbate in children with abnormal pH-MII studies, and certain glycerophospholipids were significantly higher in children not taking proton pump inhibitors (PPI) [25]. These findings highlight how reflux affects lung metabolite profiles and suggest new directions for future biomarker research on extraesophageal reflux diseases.

In contrast to a previous study, which found no significant correlation between symptoms and reflux events, our results show positive correlations between fiberoptic ENT findings, such as hyperemia of the hypopharynx and arytenoid regions, and proximal acidic and weakly acidic GER episodes. In a previous study by Simons and colleagues, they found that in 36 children who primarily suffered from dysphonia or cough, the mean RFS for those diagnosed with LPR was 5.6 [26]. Our own research confirms this, showing a similar mean RFS of 5 in children diagnosed with LPR. However, when these results are compared with those of adults diagnosed with LPR, whose mean RFS was significantly higher at 9.13 [27], it was suggested that the scoring scale used to assess endoscopic laryngeal findings in adults may not be appropriate for accurately assessing the pediatric larynx.

Our results showed interesting correlations between specific fibroendoscopic findings and different types of GER episodes. The positive associations between hypopharyngeal hyperemia, arytenoid hyperemia, and arytenoid erythema suggest that these laryngeal changes may serve as reliable indicators of the presence and severity of GER. This has important clinical implications, as fibroendoscopy could play a central role in diagnosis by providing information on the type and extent of reflux-related laryngeal damage. On the other hand, Carr and colleagues claimed that severe arytenoid edema, posterior glottic edema, and lingual tonsil enlargement in children are indicative of reflux [28]. Therefore, only arytenoid edema was found in both studies. A systematic review by Saniasiaya and Kulasegarah examined the relationship between reflux (both laryngopharyngeal and gastroesophageal) and dysphonia in children [29]. Analyzing five clinical studies with a total of 606 patients, the authors found a strong association between reflux and voice disorders, with male children accounting for 63% of the cases. Common findings on laryngoscopic examination include interarytenoid erythema and edema, erythema and edema of the vocal cords, and postglottic edema [29]. Overall, this review highlights a significant association between reflux and dysphonia in pediatric patients. In contrast, a study by Mandell and colleagues, which focused on hoarse children, found no significant correlation between endoscopic laryngeal findings and reflux [30]. The discrepancy between the laryngoscopic findings in the different studies could be due to differences in the sample size, recording techniques, and clinical profiles of the patients involved. In addition, certain laryngoscopic observations suggestive of LPR are also seen in healthy individuals, which emphasizes the difficulty of diagnosing LPR in children based on endoscopic findings alone [31].

The positive correlations observed between RSI and RFS with different types of GER episodes underline the importance of subjective symptoms and objective findings in the assessment of pediatric LPR. This interplay between patient-reported symptoms and endoscopic assessments underscores the validity of both measures and emphasizes their complementary role in providing a comprehensive understanding of the disease.

The positive correlation between the presence of eosinophils in nasal swabs and different GER episodes emphasizes the intricate relationship between upper airway inflammation and LPR. Eosinophils are known to be key players in allergic and inflammatory responses, and their presence in nasal swabs indicates localized inflammation of the upper airways [32]. Interestingly, the lack of correlation with IgE levels could indicate that this eosinophilic response is not triggered by systemic allergic reactions but by local inflammatory processes, specifically in the upper airways. This could mean that the presence of eosinophils in nasal swabs indicates a local inflammatory response to GER rather than a broader systemic allergic reaction [33]. These findings prompt a re-evaluation of the utility of eosinophils in nasal swabs as a potential biomarker for GER in pediatric patients. Rather than solely reflecting systemic allergic tendencies, eosinophils in nasal swabs could serve as a valuable indicator of local inflammation associated with LPR and provide insights into the pathophysiology and potential diagnostic markers for this condition in children.

Our study provides valuable insights into the complex landscape of pediatric LPR diagnoses. The finding that children with chronic cough and recurrent bronchial obstruction had a significantly higher number of reflux episodes than those with other symptoms suggests a strong association between these respiratory symptoms and LPR. Chronic cough and recurrent bronchial obstruction are common respiratory symptoms in children [34]. While they can be attributed to a variety of causes, including infectious, allergic, or structural problems, this study emphasizes the potential role of LPR in exacerbating or causing these symptoms. These findings underscore the importance of considering LPR as a possible etiological factor in children with chronic cough and recurrent bronchial obstruction, particularly when other common causes have been ruled out.

The correlations found between fibroendoscopic findings, eosinophils in the nasal swab, and symptom scores with GER episodes offer a multifaceted perspective on the pathophysiology of the disease. These results suggest that a holistic approach, including both clinical assessments and objective measurements, is critical for accurate diagnosis and tailored management. Understanding these relationships opens up opportunities for targeted interventions. Tailoring treatments based on specific fibroendoscopic findings or addressing localized eosinophilic responses could represent new approaches for the management of pediatric LPR. Future research should explore these therapeutic strategies and aim to improve the outcomes and quality of life of affected children.

The strengths of this study include its comprehensive approach to outcome measures, particularly its focus on a pediatric population, which fills a crucial gap in the literature. The objective monitoring techniques increase the credibility of the results, while the correlational analysis provides insights into possible associations and mechanisms.

However, this study is not without limitations. A sample size of 113 patients, while valuable, may limit the generalizability of the results. A larger sample size could provide greater statistical power and improve the robustness of the results. Another limitation to consider is the fact that the study was conducted in a single center. This design could lead to biases related to the specific patient population, practices, and protocols of the selected center. A multi-center approach could provide a broader perspective and improve the external validity of the study. While the cross-sectional design is informative, it only provides a snapshot of the relationship between symptoms and reflux episodes at a single point in time. A longitudinal study design can provide insights into temporal trends and changes over time. We recognize that there are limitations in determining the exact extent of reflux reaching the airways, and as such, our study focused on proximal reflux events as a proxy for potential airway involvement. While direct visualization of reflux into the airways is not feasible with MII-pH alone, we support this hypothesis by correlating proximal reflux episodes with fibroendoscopic findings, such as hyperemia and erythema of the arytenoid and hypopharyngeal regions, which are suggestive of LPR-related inflammation. This indirect approach, combined with symptom reports and eosinophil presence in nasal swabs, strengthens the evidence that reflux may contribute to airway inflammation, although direct confirmation of reflux in the airways would require more invasive methods, such as bronchoscopy or video-endoscopy. Furthermore, prior research has demonstrated that the proximal extent of reflux plays a critical role in the development of laryngeal symptoms. [35]. Another limitation is our reliance on the number of GER episodes as the primary measure of reflux activity. Although this parameter is valuable in identifying reflux events, the Lyon Consensus 2.0 emphasizes acid exposure time (AET) as a more robust indicator of mucosal damage [36]. Including AET in future investigations could provide a more comprehensive understanding of the pathophysiological mechanisms underlying LPR-related symptoms and enhance the clinical applicability of our findings. In addition, this study may not account for all potential confounding factors that could influence the association between symptoms and reflux episodes. Variables such as dietary habits and lifestyle could influence the results and lead to uncontrolled variations. In addition, symptom reporting is inherently subjective despite objective monitoring. Variability in symptom perception and reporting could confound the observed associations.

## 5. Conclusions

Our study provides valuable insights into the features of MII-pH in pediatric patients with suspected LPR. Among 113 children, chronic cough was the predominant symptom, followed by recurrent bronchial obstruction and throat clearing. The analysis revealed that children with chronic cough and recurrent bronchial obstruction had a significantly higher number of reflux episodes, particularly weakly acidic episodes, than those in the other symptom groups. Fibroendoscopic findings, such as hypopharyngeal hyperemia, arytenoid hyperemia, and arytenoid erythema, were positively correlated with proximal acidic and weakly acidic reflux episodes. Additionally, the presence of eosinophils in nasal swabs showed a significant positive correlation with acidic and proximal reflux episodes), suggesting localized upper airway inflammation linked to LPR. Interestingly, no positive correlation was observed with IgE levels, indicating that the eosinophilic response is likely mediated by reflux rather than by systemic allergic mechanisms. These findings reinforce the potential of eosinophils in nasal swabs as a biomarker for reflux-related upper airway inflammation in children. Our findings highlight the importance of considering LPR in the differential diagnosis of chronic cough and recurrent bronchial obstruction, especially when other common causes have been ruled out. This study underscores the utility of MII-pH monitoring in identifying weakly acidic reflux episodes and the role of fibroendoscopic findings in diagnosing and assessing the severity of reflux-related damage. Future research should explore targeted therapeutic strategies, such as interventions aimed at reducing localized eosinophilic inflammation or addressing fibroendoscopic findings, to improve the outcomes and quality of life of pediatric patients with LPR. Further longitudinal and multi-center studies, incorporating parameters such as acid exposure time (AET), are warranted to validate and expand these findings.

## Figures and Tables

**Figure 1 diagnostics-15-00034-f001:**
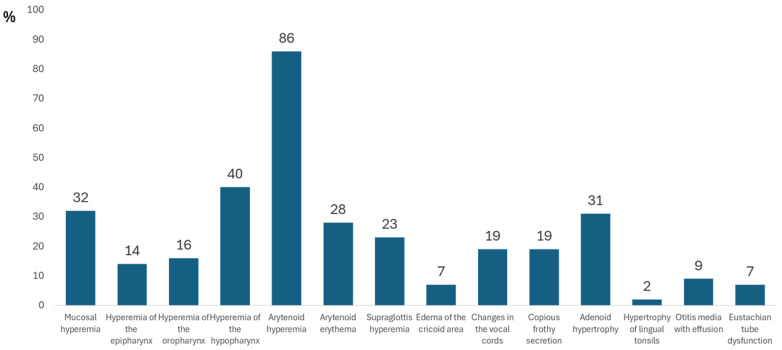
Ear, nose, and throat (ENT) findings.

**Table 1 diagnostics-15-00034-t001:** Demographic data, main symptoms, RSI, and RFS of the included patients.

	*n* = 113
Age, years (mean, SD)	8 (0.4)
Gender, male (%)	71 (63%)
Chronic cough, number (%)	53 (47%)
Recurrent bronchial obstructions, number (%)	23 (20%)
Throat clearing, number (%)	17 (15%)
Otitis media with effusion, number (%)	10 (9%)
Globus sensation, number (%)	9 (8%)
Recurrent pneumonia, number (%)	1 (1%)
Gastrointestinal symptoms, number (%)	21 (19%)
RSI (median, range)	9 (4–17)
RFS (median, range)	5 (0–10)

RSI—Reflux Symptom Index; RFS—Reflux Findings Score.

**Table 2 diagnostics-15-00034-t002:** Primary outcome measures of the included patients.

	*n* = 113
Number of reflux episodes (median, IQR)	63 (57)
Number of acidic reflux episodes (median, IQR)	25 (33)
Number of weakly acidic reflux episodes (median, IQR)	31 (36)
Number of non-acidic reflux episodes (median, IQR)	0 (1)
Number of proximal reflux episodes (median, IQR)	24 (36)
Number of proximal acidic reflux episodes (median, IQR)	13 (21)
Number of proximal weakly acidic reflux episodes (median, IQR)	9 (17)

IQR—interquartile range.

**Table 3 diagnostics-15-00034-t003:** Differences in the number of reflux episodes between the presenting symptoms.

	Chronic Cough (*n* = 53)	Recurrent Broncho-Obstructions (*n* = 23)	Throat Clearing (*n* = 17)	Chronic Otitis With Effusion (*n* = 10)	Globus Sensation (*n* = 9)	*p*-Value
Number of reflux episodes, mean (SD)	87.2 (66.2)	89.7 (45.6)	53.8 (36.1)	25.2 (13.8)	37.4 (27.2)	0.001
Number of acidic reflux episodes, mean (SD)	33.1 (21)	31.6 (28.8)	27.4 (22.6)	14.7 (12.9)	11.7 (9)	0.021
Number of weakly acidic reflux episodes, mean (SD)	48 (46.2)	49.5 (30)	23.1 (19.6)	10.5 (5.9)	23.4 (15.6)	0.004
Number of proximal reflux episodes, mean (SD)	40 (32.2)	42.1 (25.2)	22.8 (21.5)	9.5 (8.1)	11.3 (12.3)	<0.001
Number of proximal acidic reflux episodes, mean (SD)	18.6 (15.7)	20.4 (18)	13.7 (12.7)	6.7 (6.9)	4.3 (6.1)	0.011
Number of proximal weakly acidic reflux episodes, mean (SD)	18.5 (19)	18.9 (14.5)	7.5 (10.1)	2.7 (2.9)	6.9 (8.1)	0.003

**Table 4 diagnostics-15-00034-t004:** Correlation between the number of reflux episodes and Reflux Symptom Index (RSI) and Reflux Findings Score (RFS).

	RSI Correlation Coefficient (*p*-Value)	RFS Correlation Coefficient (*p*-Value)
Number of reflux episodes	0.329 (<0.001)	0.491 (<0.001)
Number of acidic reflux episodes	0.188 (0.045)	0.429 (<0.001)
Number of weakly acidic reflux episodes	0.321 (0.001)	0.350 (<0.001)
Number of non-acidic reflux episodes	−0.096 (0.309)	0.272 (0.004)
Number of proximal reflux episodes	0.226 (0.016)	0.488 (<0.001)
Number of proximal acidic reflux episodes	0.148 (0.117)	0.446 (<0.001)
Number of proximal weakly acidic reflux episodes	−0.128 (0.177)	0.388 (0.006)

**Table 5 diagnostics-15-00034-t005:** Correlation between the number of reflux episodes and eosinophils in nasal swabs and total IgE.

	Eosinophils in Nasal Swab Correlation Coefficient (*p*-Value)	IgE Correlation Coefficient (*p*-Value)
Number of reflux episodes	0.370 (<0.001)	−0.182 (0.055)
Number of acidic reflux episodes	0.360 (<0.001)	−0.162 (0.089)
Number of weakly acidic reflux episodes	0.303 (0.001)	−0.145 (0.126)
Number of non-acidic reflux episodes	0.006 (0.496)	−0.865 (0.497)
Number of proximal reflux episodes	0.341 (<0.001)	−0.232 (0.014)
Number of proximal acidic reflux episodes	0.344 (<0.001)	−0.196 (0.038)
Number of proximal weakly acidic reflux episodes	0.290 (0.002)	−0.183 (0.054)

## Data Availability

The data presented in this study are available upon request from the corresponding author.

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
