# Peer review of "Decoding Pediatric Laryngopharyngeal Reflux: Unraveling Symptoms Through Multichannel Intraluminal Impedance and pH Insights"

_diagnostics, 2024, doi:10.3390/diagnostics15010034_

Round 1
Reviewer 1 Report
Comments and Suggestions for Authors
The study was centered around pediatric laryngopharyngeal reflux (LPR), which holds considerable clinical importance. The research design was well-conceived. By employing a prospective research approach and integrating diverse techniques like multichannel intraluminal impedance - pH (MII - pH) monitoring and fiberoptic laryngoscopy, it carried out a comprehensive evaluation of children suspected of having LPR, thereby furnishing valuable data that contribute to a deeper comprehension of pediatric LPR.
The study also has some limitations, such as a relatively small sample size and the possible bias in a single-center study. Some symptoms may not be caused by laryngopharyngeal reflux.
Author Response
Dear Reviewer 1,
We would like to sincerely thank you for your time and thoughtful feedback on our manuscript titled " Decoding Pediatric Laryngopharyngeal Reflux: Unraveling Symptoms Through Multichannel Intraluminal Impedance and pH Insights".
Your comments and suggestions were highly valuable and have significantly contributed to the improvement of our work. We have carefully considered each of your points and have revised the manuscript accordingly. Below, we provide a detailed response to your comments, outlining the changes made in the manuscript and addressing your concerns.
Q1. The study was centered around pediatric laryngopharyngeal reflux (LPR), which holds considerable clinical importance. The research design was well-conceived. By employing a prospective research approach and integrating diverse techniques like multichannel intraluminal impedance - pH (MII - pH) monitoring and fiberoptic laryngoscopy, it carried out a comprehensive evaluation of children suspected of having LPR, thereby furnishing valuable data that contribute to a deeper comprehension of pediatric LPR. The study also has some limitations, such as a relatively small sample size and the possible bias in a single-center study. Some symptoms may not be caused by laryngopharyngeal reflux.
Authors: Thank you for your insightful comments on our study.
We acknowledge the limitations of our research, including the relatively small sample size and the potential bias associated with a single-center study. Regarding the concern about symptoms potentially not caused by LPR, we agree that this is an important consideration. However, one of the key motivations for our investigation was precisely this diagnostic uncertainty. In clinical practice, we often encounter pediatric patients with symptoms suggestive of LPR but lacking clear alternative explanations after thorough evaluations. Given the absence of identifiable causes for some of these symptoms, we considered it plausible that LPR might play a role and proceeded to investigate this possibility further. Our aim was to provide a more comprehensive assessment using advanced techniques, such as multichannel intraluminal impedance-pH (MII-pH) monitoring and fiberoptic laryngoscopy, to better understand the relationship between these symptoms and LPR. We hope this clarification underscores the rationale behind our study and its potential contribution to the field. We appreciate your feedback and remain open to further discussions to refine and strengthen our work.
Thank you once again for your constructive feedback. We believe the revised version of the manuscript is stronger, and we hope it now meets your expectations.
Sincerely,
Authors

Reviewer 2 Report
Comments and Suggestions for Authors
Dear colleagues!
I read with pleasure and interest your manuscript entitled "Decoding Pediatric Laryngopharyngeal Reflux: Unraveling Symptoms Through Multichannel Intraluminal Impedance and pH Insights" submitted to the Diagnostics. It is based on the data of prospective single-centre observational study aimed to assess association between fiberoptic ENT examinations, clinical signes based on reflux symptom index, molecular markers (number of eosinophils in nasal swab and serum IgE) with number of gastroesophageal refluxes of different types (acidic, weak-acidic, non-acidic) and their proximal extent. This study may bring new to the field and might be interesting to the readers. I have a few minor comments described below.
First, the study group seems to be heterogeneous and this may make it's results non-reproducible and therefore poorly eligible for publication (at the present state). You did not provide reasons why symptoms and conditions described in lines 88-91 were supposed to be associated with gastroesophageal reflux. Actually, globus sensation is rarely symptom of GERD. I suppose that some diagnostic efforts had been taken to exclude other aetiologies of such conditions like chronic cough, recurrent bronchial obstruction and throat clearing. Whether all the confounders were excluded (dust? dry air? smoking? medications? etc). Please, describe this diagnostic workup in details.
It seems strange that among 113 children there was no one experiencing heartburn / acid regurgitation, as indicated in lines 99-100. Was it an exclusion criterion?
Please, describe diagnostic methods as it is normally required: provide details of fiber optic laryngoscopy device (model/type/manufacturer), whether video recording or endo-photo documentation was performed; whether automatic analysis of MII-pH tracings were used or the data were checked manually (which may be important, especially for Laborie equipment), whether meal periods were excluded from the analysis; which of the laryngopharyngeal symptoms where checked for the timing of occurrence and wether (if yes - how) their association with GER were analyzed. In terms of laboratory data - please, provide the details on timing of samples collection, further processing and laboratory methods (name of method, equipment, reagents) used.
Detection of proximal GER: is it possible to be sure that the extent of GER was proximal enough to get into the airways?
Outcome measures: number of GERs is supposed "supportive" for GERD diagnosis according to the Lyon 2.0, whereas acid exposure time is more important in terms of mucosa damage not only in the esophagus, but also in airways. I suppose this should be at least mentioned in the discussion.
Section of conclusions: could you please shortly summarize the data of you study there, with support of the digital data obtained - what was found in what group and (optional) what it means for practice/further research.
Minor note: as far as I know "weakly alkaline" refluxes are nowadays supposed "non-acid".
Please, modify the tables according to general rules: tab 1 and 2 - with name of the parameter mentioned at the headIng (first line); please use "IQR" instead of "range" (as min-max is also a range). Tab 4 - please, merge upper cells in columns 2-3 and 4-5 so that RSI and RFS were above (it is possible to merge cells for each parameter) and "correlation coefficient" and "p-value" were below; use R or "correlatIon coefficient" instead of just a "coefficient" (which is not clear). Similar measures should be taken for tab. 5.
Fig. 1 - chart bars names and values are poorly visible.
Self-citation exceeds 10% of the references.
In general, I suppose that the mentioned flaws may be easily corrected and hope that my comments help you to make the manuscript even better.
Comments on the Quality of English LanguageSome typos are detected, minor language polishing would be appreciated.
Author Response
Dear Reviewer 2,
We would like to sincerely thank you for your time and thoughtful feedback on our manuscript titled " Decoding Pediatric Laryngopharyngeal Reflux: Unraveling Symptoms Through Multichannel Intraluminal Impedance and pH Insights".
Your comments and suggestions were highly valuable and have significantly contributed to the improvement of our work. We have carefully considered each of your points and have revised the manuscript accordingly. Below, we provide a detailed response to your comments, outlining the changes made in the manuscript and addressing your concerns.
Thank you once again for your constructive feedback. We believe the revised version of the manuscript is stronger, and we hope it now meets your expectations.
Sincerely,
Authors

Round 2
Reviewer 2 Report
Comments and Suggestions for Authors
Dear colleagues!
I read both, the revised version of the manuscript and the Authors' reply to reviewer's comments. I can conclude that most of my comments were addressed, the manuscript was significantly improved and may be accepted for publication.
Minor notes: Tab 2 - please, mark "range" with asterix (*) in the column 1 cells and transfer "IQR – interquartile range" from the table's title to the footnote or use "IQR" in the cells.
Table 3 - p-value is a nice option... But is it really necessary in this particular table (as it is not clear what it really reflects - heterogeneity of reflux-associated factor impact on reflux-associated symptoms?)?
These remarks relate only to the quality of data presentation and in no way diminish the scientific value of the work itself.
Author Response
Dear Reviewer,
Thank you very much for your careful evaluation of our revised manuscript and for acknowledging the improvements made. We are grateful for your valuable suggestions regarding the quality of data presentation, and we have addressed your comments as follows:
Q1. Tab 2 - please, mark "range" with asterix (*) in the column 1 cells and transfer "IQR – interquartile range" from the table's title to the footnote or use "IQR" in the cells.
Authors: Thank you. Changed as suggested.
Q2. Table 3 - p-value is a nice option... But is it really necessary in this particular table (as it is not clear what it really reflects - heterogeneity of reflux-associated factor impact on reflux-associated symptoms?)?
Authors: Thank you. This p-value indicates that there are differences in the number of different reflux episodes and groups of patients stratified by symptoms. This is now added in the description of the table.
We sincerely thank you for your constructive feedback, which has helped improve the quality and presentation of our work. We greatly appreciate your positive assessment.
Best regards,
Authors